# The Frontline Nurse’s Experience of Nursing Outlier Patients

**DOI:** 10.3390/ijerph17145232

**Published:** 2020-07-20

**Authors:** Jasmine Cheung, Sandra West, Maureen Boughton

**Affiliations:** 1School of Nursing, Tung Wah College, Hong Kong, China; 2Susan Wakil School of Nursing and Midwifery, University of Sydney, NSW 2006, Australia; sandra.west@sydney.edu.au (S.W.); maureen.boughton@sydney.edu.au (M.B.)

**Keywords:** nursing the outlier patients, nursing specialty, specialized nurse, professional nursing practice, nursing competency, hermeneutic phenomenology

## Abstract

The frontline nurses’ experience of nursing with overstretched resources in acute care setting can affect their health and well-being. Little is known about the experience of registered nurses faced with the care of a patient outside their area of expertise. The aim of this paper is to explore the phenomenon of nursing the outlier patient, when patients are nursed in a ward that is not specifically developed to deal with the major clinical diagnosis involved (e.g., renal patient in gynecology ward). Using a hermeneutic phenomenological approach, eleven individual face-to-face in-depth interviews were conducted with registered nurses in New South Wales, Australia. The study identified that each nurse had a specialty construct developed from nursing in a specialized environment. Each nurse had normalized the experience of specialty nursing and had developed a way of thinking and practicing theorized as a “care ladder”. By grouping and analyzing various “care ladders” together, the nursing capacities common to nurses formed the phenomenological orientation, namely “the composite care ladder”. Compared to nursing specialty-appropriate patients, nursing the outlier patient caused disruption of the care ladder, with some nurses becoming less capable as they were nursing the outlier patient. Nursing the outlier patient disrupted the nurses’ normalized constructs of nursing. This study suggests that nursing patients in specialty-appropriate wards will improve patient outcomes and reduce impacts on the nurses’ morale.

## 1. Introduction

The development of complex, modern hospitals has created wards/units specific to medical or surgical specialties. One of the advantages of this co-location of patients with a similar clinical diagnosis is the development, overtime, of a body of shared knowledge and practice by the registered nurses attached to that ward. However, optimal bed utilization means that nursing the outlier patient is and will continue to be a common experience for many frontline nurses in acute care settings [1,2,3]. Nursing patients in a ward that is not addressing the specific needs of the outlier patients has implications on the nurses’ professional roles and responsibilities, consequentially affecting patients’ health outcome and nurses’ morale [4,5]. In this paper, findings from a hermeneutic phenomenological study conducted in New South Wales, Australia, are reported. This paper sheds light on a phenomenon that has remained largely unsaid and unexplored from the nurse’s perspective [6].

## 2. Background

The phenomenon of nursing the outlier patient has been underreported as nursing with visible consequence because of hospital bed shortages in Australia [5]. To many nurses, outlier patients are human beings needing to be nursed in a ward that is not suited or equipped to address their specific needs, rather than the positioning of such patients as a consequence of political actions. Theoretically, all nurses possess a certain degree of knowledge and skills to care for the patients. The competent nurses, along with the “unavoidable” occurrence of outlier patients, outline the scene for exploring the studied phenomenon. Regardless of patients being outlier patients or specialty-appropriate patients, all patients deserve a certain “standard of care”, and all nurses owe a “duty of care”.

### 2.1. The Competent Nurses

Clinical competence of a nurse, as described by Benner, encompasses five levels of proficiency, namely “novice, advanced beginner, competent, proficient, and expert” [7]. Rather than relying on the nurse’s age or years of experience, the acquisition of proficiency and the development of the nurse’s individualized care perceptions and practice depend on the nurse’s exposure and integration of “actual practical situations” with the scientific research [7,8]. However, the rarely occurred “actual practical situations”, such as nursing patients with rare diseases or applied equipment, tend to foster competent nurses at the novice level. Unlike the case of nursing for patients with rare disease, the studied phenomenon goes beyond the consideration of the health professionals’ lack of knowledge for all diseases or rare disease process [9]. The similarities between the occurrence of patients with rare diseases and that of outlier patients comprise the minority in the acute care setting. Conversely, effective treatments or evidence-based management guidelines for outlier patients may be present but not as much for patients with rare diseases [10]. Expectation for nurses to have the specialty knowledge and skills specifically for caring for patients with rare diseases may be unrealistic. However, the occurrence of outlier patients itself is not a rare event and may result in expectations similar to those of specialty-appropriate patients [2].

### 2.2. The “Unavoidable” Occurrence of Outlier Patients 

Historically, a hospital was a boundaryless place where patients who were admitted with various conditions were grouped together anywhere under the same roof, for example, in the general medical ward or in the surgical ward [11]. With the rapid development of scientific knowledge and evolving economic complexity, a hospital has changed from an open-ward-area to multiple wards that are organized to cater for a variety of new specialties along with the development (or creation) of new units [12,13]. The delineation of ward boundaries sets the background for the occurrence of outlier patients, being nursed away from the specialty-appropriate wards. 

Similar to other developed countries, public hospitals in Australia have been facing the challenges of limited resources and increasing demands to improve the quality of patients’ outcomes [14]. Despite the stated health system’s aims to “provide the right treatment in the right place within an appropriate timeframe every time for every patient” [15], occurrence of outlier patients has been strategically designed to maximize bed utilization in times of overstretched hospital resources. Australia has a lower number of hospital beds (4.0 beds per 1000 members of the population) in public and private hospitals than the average of Organization for Economic Co-operation and Development Countries and other selected countries (4.8 beds per 1000) [16]. Moreover, Australia is facing an increasing demand from an ageing population, increasing levels of chronic disease and limited access to services in certain regions [17]. While hospital acute bed occupancy data for Australia were unavailable [14], it is possible that Australia has an excessive bed occupancy due to the limited hospital beds and high demands from the population for healthcare services. It is believed that Australia’s bed occupancy rate is at the upper end of comparable countries. Recent figures show that in Hong Kong, the medical inpatient bed occupancy rate was 121% [18], whereas England’s average occupancy rate for general and acute beds overnight was 90.2% [19]. Unsafe work environment, from an infection control perspective, results when the average bed occupancy exceeds 82–85% [20]. 

As a result of high demands for hospital beds, inpatient bed utilization has been pushed by the “hospital’s overriding concern about the excessive spare bed capacity” [21]. Patient misplacement has resulted as most bed planning queueing models attempt to overcome bed shortages [22]. As hospitals continue to accept admissions when bed demand is higher than bed supply within a particular ward, patients of a particular condition will end up in a mismatching ward and become an outlier patient [23]; since the last decade, outlier patients were being identified as a “problem” in the acute care settings in New South Wales: “An associated problem occurs where patients end up in the wrong bed, that is, in a ward inappropriate for their condition simply because of the unavailability of a more appropriate bed. These patients are called―outliers.” [5] (p. 990).

Recent literature suggested that misplacement of outlier patients creates a higher risk of complications than specialty-appropriate ward [23,24]. Since all healthcare holds inherent risks, minimizing the risks to patients becomes an important aspect of practice [25]. Advancements in bed usage technology have made the presence of outlier patients visible as statistical variance for care coordination and task management purposes. In New South Wales, the Patient Flow Portal Bed Board can be used to distinguish between patients who need short-term clinical support in another unit (e.g., aged care patients who need short-term clinical support in another unit such as aged care patients needing 24 h of monitoring in a cardiac ward) and the true outlier patient who is placed in an available but inappropriate bed [15]. One justification of these placements is that the remaining outlier patients have low risk of clinical deterioration and are being considered as relatively safe to be allocated to any hospital location. 

### 2.3. Nursing With Lower Level of “Duty of Care” 

The uniqueness of nursing practice relies on the nurse’s ability to respond to the changing healthcare system and to the nursing needs of each individual in their care [25]. This unique practice must be available, “wherever there is a patient in need of care” [26]. In Australia, nursing practice involves applying skills and knowledge to ensure safe and effective delivery of health services in nursing [27]. Nurses are legally bound to be accountable for their “decisions actions, behaviors and responsibilities” [28]. This inclusiveness of all patients in nursing practice indicates that nurses have a “duty of care”, regardless of whether they are nursing outlier patients or specialty-appropriate patients. The nurse is responsible to “know who the outliers are and what medical teams are involved” and to raise any safety concerns that require immediate action [29]. Traditionally, nurses have a higher level of duty of care when compared to other carers with limited or no medical knowledge [30]. However, the maintenance of this high level of “duty of care” in nursing the outlier patient became questionable as nurses “were not experienced or have less experience in looking after” outlier patients when compared to nursing specialty-appropriate patients [31]. A study taken from the outlier patient’s perspective reported that nurses nursing outlier patients had “compromised knowledge” at times and were “unfamiliar with the nursing care required” with “minor mistakes” being made [32]. These societal expectations and perceptions of nursing form part of the nurse’s experience of nursing in the acute care setting.

### 2.4. Nursing With Lower Level of “Standard of Care”

Outlier patients were less likely to receive the “same standard of care” as specialty-appropriate patients [24,33]. Patients, clinical staff and administrators perceived that the presence of outlier patients may lead to safety concerns [34]. Outlier patients are also associated with more emergency calls, more comorbidities, longer length of stay and higher readmission rates when compared to specialty-appropriate patients [2,24,35]. Most current research focuses on reporting the consequences of outlier patients, rather than issues related to nursing staff. 

In Australia, processes for recognizing specialty areas within nursing have been well developed, and the healthcare system is at “an appropriate level of public protection, whist ensuring a dynamic, flexible and responsive workforce” [36]. Despite research documenting a foreseeable lower level of “Duty of Care” and “Standard of Care” when comparing to nursing specialty-appropriate patients, the nurse is expected to “take action” in response to the “unavoidable” occurrence of outlier patients [32]. Accepting responsibility and accountability for the nurses’ own practice is one of the important professional values held by nurses. The occurrence of outlier patients causes “a source of stress” for nurses and “extend(s) nurses beyond their normal areas of expertise” [37]. Insufficient beds in the acute care setting further increases the workload and leads to poor nursing morale [5]. Rather than prematurely dismissing the phenomenon as an avoidable ethical challenge and moral stress [38,39], this study sheds light on the nurse’s experience of nursing outlier patients. While studies regarding the occurrence of outlier patients from patients’ and administrators’ perspectives have been completed [32,34], scant studies are available to explore the studied phenomenon from the frontline nurses’ perspectives. The purpose of this phenomenological inquiry is to develop an understanding of the nurse’s experience of nursing the outlier patients, in other words, the nurse’s experience of fitting care for patients whose illness conditions are a misfit with the ward specialty. 

## 3. Methods

Unstructured, face-to-face in-depth interviews were conducted with nurses to explore their experience of the studied phenomenon. The Consolidated criteria for Reporting Qualitative research Checklist (COREQ), as shown in Appendix A, was adopted to illustrate a clear audit trail [40].

### 3.1. Research Design

The study used a hermeneutic phenomenological approach, as informed by Heidegger’s Being and Time [41], together with Merleau-Ponty’s [42] and van Manen’s [43] discussion on temporality and spatiality. Hermeneutic phenomenology is a qualitative approach focusing on the process of understanding and interpreting human experience [43]. Nursing the outlier patient is not an event occurring at any one point of the clock time. As the nurse is nursing, she experiences and perceives nursing. It is this lived experience and perception of nursing that serves as the basis for later interpretation of nurses’ experience of nursing outlier patients. Moving beyond the ontological description of the relatedness between the corporeal presence of humans and their experience that shapes their Being-in-the world [41], hermeneutic phenomenology allows the fusing of participant’s experience, together with the researcher’s interpretation as a nurse and as a researcher. This “fusion of horizons” facilitates new understandings of phenomena [44]. 

### 3.2. Criteria for Participant Selection and Recruitment 

Eleven registered nurses, who had been working in a public hospital in New South Wales within the past 2 years before commencement date of the study and who had the experience of nursing outlier patients were recruited to participate in face-to-face in-depth interviews. Participants were mainly recruited through an advertisement published in the Lamp, the magazine of the New South Wales Nurses’ Association. The nonprobability sampling techniques, including purposeful sampling and snowball sampling, were adopted. 

The first contact with participants was made by the first author via telephone prior to conducting the interview with them. Each participant was given a copy of the participants’ information statement to ensure that they understood the purpose and the procedures required for the project. Written informed consent was obtained from all participants prior to interviewing. It was explained to the participants that the interview was completely voluntary and that they could withdraw from the study at any time without penalty. 

### 3.3. Ethics

Ethical approval was granted by the University of Sydney Human Research Ethics Committee (Ref No.: 03-2009/11602), Australia. Written informed consent was obtained from the eleven participants prior to interviewing. Interviews were conducted at various locations nominated by the participants as comfortable environments that also met the researcher safety requirements of the ethics protocol..

### 3.4. Interview Procedure

In order to allow maximum freedom for participants in reporting their experience, a recursive model of interviewing was used [45]―“*Can you tell me about your experience of nursing outlier patients*?” was the only question posed for participants at the beginning of the interview. The natural flow of conversation was then guided by the participants’ responses. Probing questions were asked in order to clarify information or to enhance depth of a response [44]. All the interviews were conducted by the first author, who was a PhD student at the time of interview. The interviews ranged from thirty minutes to one and a half hours. 

All interviews were audio recorded. Data collection period ranges from June 2009–January 2011. Table 1 presents the demographic characteristics of the participants.

### 3.5. Analysis

The researcher used her “personal experience as a starting point” [43]. Summaries and reflections were written by the first author immediately after each interview, during interview transcription and after conduction of line-by-line analysis for each transcript. The first author referred to her written summaries and reflections while seeking meaning and uncovering initial thematic aspects [46]. Analysis was conducted through a process of interpreting participant’s interview transcripts initially word by word and then sentences and ideas until significant aspects of their experience were determined and clustered subthemes merged into themes. The Scrivener software program (Literature & Latte, Cornwall, United Kingdom), was used to record all the sub-themes as note cards on the cork-board electronically. The note cards sharing commonalities, in other words, the sub-themes in common were then visually put into a file. Each file represented one of the five themes in this study. Further probing of subthemes and themes led to the emergence of the “phenomenological orientation” for later phenomenological analysis [43]. Constant comparisons were made between the first author’s analysis and the independent analysis conducted by second and third authors [45]. The issue of rigor was best illustrated by van Manen, where he says that phenomenological description is “validated by lived experience and it validates lived experience” [43].

The audit trail in Table 2 below demonstrates the non-linear process of phenomenological data analysis applied in data analysis. Table 3 further provides an example of the conduction of initial thematic analysis for this study. 

## 4. Results

Phenomenological orientation serves as an important reference for exploring the temporal and spatial features of the studied phenomenon. In relation to this study, nurses nurse in clock time and in ward place, yet experience lived time and lived space. Lived time is known as temporality [39]. Unlike the linear clock time of past, now/present and future, temporality is constituted of three interchangeable horizons, including “the horizon of the past, horizon of imminent and horizon of futurity” [42]. Temporality is defined by the connectedness among horizons [42,47,48]. For instance, nurses may learn from the past and anticipate the future, during their nursing at present. Alternatively, nurses may stay in the past and remain nursing in a fixed routine at present, without anticipating the future. Similar to lived time, lived space moves beyond the ontological definition of architectural space or setting arrangement. Lived space can only be experienced when the space “affect(s) the way we feel” [43]. An inquiry into nursing the outlier patient lies in the lived space, where the nurse experiences a difference between nursing patients in specialty-appropriate wards and nursing outlier patients in an inappropriate space for their conditions/diagnosis. The subjective experience of an individual is meaningless unless it can be understood by others. Phenomenological reflection requires transcending through temporality and spatiality. The reflective process is “painful, difficult, disorientating…like writing in the dark” [49]. Objects in daily life as described by participants can serve as a medium for engaging researchers and the audiences, as well as to support the reflective process. 

Reflecting on participants’ excerpts, each participant discussed their own capabilities and the different levels of care involved in nursing specialty-appropriate patients and in nursing the outlier patients. Participants’ description of the different levels of care can be seen in the initial themes of perceived care ladder and the initial subthemes as earlier listed on Table 3. The combination of the participants’ comments has further led me to orientate the phenomenon through visualizing it with the metaphor of a ladder, namely the “care ladder”.
“*… management just don’t care that is important to get patient back to their own area specialty … And if they are in a bed, that’s what all it matters. Doesn’t really matter we are all general nurses at the end of the day basically. And if it’s the level of care that you want for your patients in your hospital, that’s fine. But we tend to think these days and age that specialty care is far more important… Because it reduces some hospital length of stay and also increases the patients’ outcome.*”(Peter 118–124).
“*I hope that basic nursing is carried out everywhere that everyone has the best interest of the patient at heart so that they are getting adequate care … it’s the special skills that nurses required working in specialized **care** that they (the outlier patients) probably missed out on.*”(Ann 61–64).
“*It’s frustrating … you feel bad because you are not able to provide **good** nursing care. You are only able to provide basic nursing care (to outlier patients),*”(Madeline 133–134).
“*I would say appropriate is one level up from adequate care. Adequate is just that...you know you give enough … on a ladder … all different levels. You would call basic is the basics… you just do the basics... Keep them alive. Hope that they don’t get any worse. But that is just a minimum. Adequate level is already much better because it encompasses more. You have a much better picture of the whole person. Appropriate care would be really looking at all their needs where it goes to the really high level of nursing care, I would call that first class of nursing care...*”(Agnes 200–209).

Among the different forms of appearance and functions of ladder available, a simple straight ladder that supports the individual participant’s approach to nursing will be used to orientate the phenomenon. Ontologically, a simple straight ladder composes of two ladder feet, two ladder sides and set of rungs. A simple straight ladder leans on the surfaces of wall and floor. The leaning surfaces and the ladder itself are separate entities. Its function is for people to reach certain height, in other words, to archive a certain level of care. The best positioning on the ladder depends on the height that a person needs to reach. Reflecting and immersing in the data, the participants’ experience is thematically named as “a care ladder”. The care ladder serves as a phenomenological orientation for understanding the nurse’s experience of nursing. The studied phenomenon is later illustrated by the disruption of this care ladder. 

The construction process of the nurse’s care ladder is originally distinctive for each patient, although sharing some similarities among patients. At the beginning, each nurse constructs “a care ladder” for each patient. The nurse determines the capabilities required for each patient and builds a care ladder for each patient from scratch (Figure 1).

The construction process continues as the nurse then groups her experience of nursing with different care ladders to construct “the composite care ladder” as shown in Figure 2.

The composite care ladder then serves as a foundation for constructing a distinctive care ladder to meet an individual patient’s care requirements (Figure 3). Nursing with the composite care ladder has advantages over developing a new care ladder from scratch. One advantage is that the composite care ladder fits nursing for the majority of patients with a similar illness condition in the specialty ward. Another advantage is that the composite care ladder allows the nurse to respond quickly when complications occur and hence improves patient outcomes. The nurse has normalized her experience of nursing with the composite care ladder by adding or removing existing nursing capabilities (ladder rungs) regardless of nursing specialty-appropriate patients or nursing outlier patients. The construction process of care ladders for different patients are shown in Figure 3 below.

The structure of a care ladder is similar to the simple straight ladder used in everyday life, except leaning surfaces are inclusive as part of a care ladder. Surface of anchorage is a term used to describe the leaning surfaces of a care ladder. The vertical and horizontal surfaces of anchorage are interchangeable and characterized by the external world of nursing and by the external world of organization. The nurse’s external world of nursing is informed by the nurse’s past exposure to nursing, such as their experience during their preregistration education and their experience of nursing in acute care wards as registered nurses. The nurse’s experience of nursing the outlier patients refers to the formative relations between who I am and who I may become, between how I think or feel and how I act… [35] (p. 26). Comparatively, the external world of organization constructs the workplace, work and time to ensure work effectiveness and efficiency under resource constraints, hence underlying the occurrence of outlier patients. While these interchangeable surfaces on their own describe the occurrence of outlier patients, they are insufficient to inform the phenomenon. 

The nurse’s positioning of a care ladder between these surfaces has implications on the nurse’s climbing effort. The steeper the slope of the care ladder placed between these two surfaces, the more climbing effort is required from the nurses to maintain the patient’s outcome. When the climbing effort exceeds the nurse’s ability, patients’ outcome may be compromised. 

Another implication on the nurse’s climbing effort is the height of the care ladder. The height of the care ladder symbolizes the nurse’s perceived best level of care based on her previous experience of nursing, together with her perception of acuity and severity of the patient’s disease/illness condition. The height of the care ladder is dependent on the number of rungs available, where the rungs represent the nurse’s capabilities required by each patient’s conditions. The more capable the nurse is, the less effort is required for the nurse to climb the ladder. Figure 4 presents an individual nurse’s care ladder that has six rungs for the purpose of demonstration, compared with four potential rung disruptions described by the individual participant’s that represent their experience of nursing the outlier patients. Some nurses were becoming less capable of:(a)Synchronizing nursing rhythms;(b)Practicing with disease and/or condition specific familiarity;(c)Prioritizing each nursing task;(d)Predicting care requirements;(e)Practicing with inter-professional relationality, as they were nursing the outlier patient, as a result/outcome of nursing the outlier patients.

### 4.1. Becoming Less Capable of Synchronizing Nursing Rhythms 

Nursing the outlier patients requires the nurse to incorporate a different nursing rhythm into the normalized nursing routine of nursing the specialty-appropriate patients. The following explanation from John and Rainbow illustrates the desynchronized rhythm of nursing the outlier patient and that of specialty appropriate patient.
“*If that (outlier patient’s specialty) team of doctors used to do their round at 10am (at the team doctor’s specialty ward) because this suits the rest of their day, they will come to see the outliers at two o’clock, for example. Now you are trying to feed the patients. The doctors now want to see the patient or whatever. So, it can affect the nurses’ routine as well. It has to affect the patients...might affect showering, may affect the feeding, may affect the medications, may affect the dressings, it may affect the treatment times because doctors are coming on a different routine to see outliers as they come in as a normal (specialty) ward. So that now the nursing routine, everything to do with patient care (is disrupted) …*”(John 44–54).
“*… by the time you got to contact them (the clinical nurse educator (CNE) or clinical nurse specialist (CNC) for the outlier patients), there is a delay because you always do that (call the CNE or CNC) once you finish with the routine, like giving medications and up to showers or doing other things ... and then by the time the CNE or CNC comes, they may be in a meeting, so...they cannot come on time ... sometimes they are in the middle of (doing) something too ...*”(Rainbow 145–150).

Both John’s and Rainbow’s experiences of nursing specialty-appropriate patient with synchronized rhythm have been disrupted by their experiences of nursing outlier patients with de-synchronized rhythms. John is capable of fitting either the organization-scheduled mealtime or the consultation time from the healthcare team specific for outlier patients as scheduled. John becomes less capable of fixing the conflict between the two in his nursing schedule. Comparatively, Rainbow is not able to complete the dressing without contacting the CNE or CNC from the specialty of the outlier patient. The nursing activities for outlier patients are postponed until Rainbow finishes other nursing activities required by all patients. 

Another example of the desynchronized nursing rhythm is the nurses’ extra effort and time searching for equipment to care for the outlier patients. Nursing the outlier patient often requires specific equipment from another ward since the work environment is designed for patients with specific illness conditions. Hope normalizes her experience of nursing specialty-appropriate patients, in which the nursing equipment is almost always readily available in the ward. When Hope nurses the outlier patient, she finds herself stretching her work environment from one ward space to another ward space to look for the equipment. Hope uses extra energy to locate as well as use the unfamiliar equipment for that particular outlier patient. Similarly, John reports that more time is involved in getting equipment from a ward of a different specialty.
“*… (The outlier patient) has specific drains for gastro, for after the operation, but that was not on (our) ward. So that means you have to get that from the other ward, the gastro surgical ward, which is not a big job to get it, but it all added little stressors to getting the job done and looking after the patient.*”(Hope 156–159).
“*You know all wards … (are in a) very specialized world now ... the departments are completely separate, particularly true for departments of medicine and administration and budget and everything. They wouldn’t necessarily supply equipment from their budget ...*”(John 29–33).
“*…When it (goes beyond) our quota of equipment (when our ward does not have that particular equipment in stock), it takes longer. Sometimes your mate (nurses in other ward) refuses (to provide) it (for you), so then you go to find another avenue (ward) to get the equipment. And if you are looking for an equipment, (it is) not for fun, it’s part of patient care. So, its costs, kind of affects the patients. He may be waiting for that longer, when he needed that straight away.*”(John 217–220).

The timeliness of nursing is being affected, as nurses are increasingly involved with indirect patient care activities in nursing the outlier patients. The coordination of care needed for the outlier patients disturbs the nurses’ usual rhythm of nursing specialty-appropriate patients. As the rhythm of nursing the outlier patient and that of specialty-appropriate patients are partially or completely out of synchrony, the care ladder rung will be fragile or broken.

### 4.2. Becoming Less Capable of Practicing With Disease And/Or Condition Specific Familiarity

As some nurses considered an outlier patient as one who “is under another specialty”, individual participants have drawn a boundary between knowing “my specialty”, and not knowing outside of “my specialty”.
“*If it (the outlier patient) is surgical there is a lot of ... drains that you know really little about. And if something goes wrong with any of those drains it’s not easy.*”(Rainbow 61–63).
“*...there is a lot of this tubing and ...containers and drains...and you don’t even know the names of them. Like they are saying Bellovac. There are different (unfamiliar) names.*”(Rainbow 80–82).
“*I think I have done everything for this (outlier) patient but I haven’t been able to educate them as much as I normally would if it is a cardiac condition ... It’s just that maybe I could (have) done a bit better…You don‘t know what you don‘t know really.*”(Marie 57–60, 292–293).

While Rainbow developed familiarity from experience with pericardial drains as a cardiac nurse, she did not know how to operate the Bellovac drain. Bellovac drain is a drain with not only a different name but also located in a different part of the body with unknown conditions of use and potential risks. Her familiarity in specialty cardiac nursing has not been transferrable to nursing the patients from other specialties. Similarly, Marie expressed her lack of or decreased capability to educate outlier patients due to her diminished familiarity, which consequentially leads to uncertainty. She is unable to use her cardiac-related knowledge and knowing to educate outlier patients, suggesting that the knowledge and knowing between two different specialties may only be minimally linked. The lack of familiarity in the nurse’s practice of nursing the outlier patient is represented by the absence of the care ladder rung.

### 4.3. Becoming Less Capable of Prioritizing Each Nursing Task 

Nursing the outlier patient leads to nurses not being able to prioritize nursing care requirements or be decisive about appropriate care implementation.

The participants experienced frustration at the perceived additional amount of time that outlier patients consumed in their busy schedule compared to nursing their specialty-appropriate patients. According to Hope, nursing the outlier patient has taken her time away from nursing specialty-appropriate patients so that all of the patients in her care are affected by the one outlier patient.
“*So I had the added stress of not being able to look after the other patients as well as I should have done ... I think everybody was missing out. I was stressed. But the patient who was in the wrong ward, he (the outlier patient) really suffers more because he was in the wrong ward. And the other patient (specialty-appropriate patients) didn’t get the care and the attention they needed. So it was very stressful and dissatisfy(ing).*”(Hope 76–80).
“*…When I mentioned that to the doctor, can we not get this patient to High Dependency Unit? Do you think he should be in a different ward? Of course, he totally agreed. When I said, will you do something about it, he just sort of shook his shoulder and left. And when I mentioned that to the team leader, she fully agreed and also did nothing. So, I mention it a few times to her. I mentioned it to the evening shift who took over from me. So yes, I don’t know, maybe I should have done more. Maybe I should have found the nursing supervisor myself, yes. Which again, there is not enough time for sitting on the phone and I really have to look after the patients. So that’s another part of being very dissatisfied and frustrated after days like that.*”(Hope 335–344).

From Hope’s experience each nursing task for the outlier patient meant sacrificing to some extent each nursing task for the other patients. Hope described “sitting on the phone (for the outlier patients)” and while she is doing this, she is not able to “… really look after the patients”. Hope had become frustrated by the additional time she had to spend making telephone calls to establish contact with different personnel who do not normally have patients in her ward and to locate necessary equipment that is outside her specialty ward. Nursing the outlier patients adds extra indirect nursing activities and requires additional time, and therefore, the nurse becomes less capable of prioritizing each nursing task. With more indirect care activities involved in nursing the outlier patients, the nurse is less capable of prioritizing, and this adds to the time constraints she is already experiencing. These added work and time constraints is represented by the distanced rungs, where more climbing effort and energy is required for overcoming the indirect care for nursing the outlier patients.

### 4.4. Becoming Less Capable of Predicting

Unlike nursing the specialty-appropriate patients where the nurse is capable of predicting future events, of being proactive and planning care ahead, both Rainbow and Claire reported a lack of confidence and the associated emotional response with caring for the outlier with the same capability.
“*You don’t feel confident doing it (nursing the outlier patients), and there is no element of predictability, you cannot determine what is going to happen next. You can’t look forward. You are only looking at what (is happening now).*”(Rainbow 161–171).
“*Well, you feel a bit anxious and stressed … what if something happens ... I might not (have) recognized or there might be something wrong … What if I don’t know the drugs (for this unfamiliar condition/diagnosis of the outlier patients), if I am not sure? Because we are used to knowing everything (about our regular/specialty-appropriate/familiar patients).*”(Claire 705–709).

When Rainbow and Claire nurses the outlier patients, they were unable to anticipate care. Claire attempted to recall from her practical familiarity or background practice of nursing specialty patients. The nurse’s incapability of predicting is represented by the absence of rungs in the phenomenological analysis, where the nurse is unable to grasp the temporal nature of their experience and to stretch themselves to future planning for patient care—hence rendering the nurse’s best level of care unreachable or distanced on the care ladder—and that requires more climbing effort and energy from the nurse.

### 4.5. Becoming Less Capable of Practicing With Inter-Professional Relationality

The difference in the rapport developed among staff in nursing specialty-appropriate patients and nursing the outlier patient was reported by Ann and Madeline.
“*So, when we have outliers, we don’t know what those plans are because we are not there with that treating team. Similarly when we have outliers in the other wards, we are really not sure what’s going on. So I see the main disadvantage is the lack of continuity.*”(Ann 27–30).
“*If it is a (other) specialty areas, for example, renal or cardiology, you are not seeing these people (allied health care members or doctors of the outlier patient’s team) everyday every week ... you don’t establish the same sort of rapport.*”(Madeline 104–105).

Both Ann and Madeline normalized their experience with the health care professionals of their specialty-appropriate patients, as they often saw them in the ward and developed more face-to-face interactions. Both pointed to a lack of shared understanding of the outlier patient’s need for care with the healthcare professional of outlier patients. As they nursed outlier patients, they could no longer link their space and the space of the health care professionals of the outlier patient in this absence of face-to-face interactions, hence deepening the absence of rapport. The absence of rung indicates that the nurse is nursing out of space and out of time as she nurses outlier patients.

## 5. Discussion

The studied phenomenon, nursing the outlier patient, has been explored phenomenologically by the initial construction of the care ladder and later disruption of the care ladder. Nursing on the care ladder with broken rung, fragile rung, absence of rung, and/or distanced rungs means that nurses are using extra effort and energy, with a prolonged timeframe to reach the nurse’s best level of care as she nurses the outlier patients. The results indicated that the presence of outlier patients is not merely a bed management mismatch of patient numbers, patient types and beds [5] but is an indication of inappropriate care. In this study, the definition of inappropriate health care as “health care risks exceed(ing) the benefits” [50], has been redefined as the nurse’s experience of not being able to provide the Duty of Care and maintain the Standard of Care outside the scope of professional nursing practice. A clearly articulated scope of practice is overdue in Australia and is necessary for ensuring the full utilization of nurses’ skills based on their education, experience and expertise in patient care [51,52]. The finding is significant as it provides the composite care ladder to illustrate the scope of practice at an individual level as informed by the frontline nurses’ experience of nursing in the acute care setting. 

### 5.1. Implications on Nursing Practice: Organizational Commitment of Nurses and the Associated Liability and Accountability

The significance of this study also lies in the debate of negligence liability when the nurses’ capabilities have been disrupted by the bed management. Participants experience a dichotomy between the practice of nursing and the theory of nursing as they are nursing outlier patients. In terms of nursing practice, our findings indicated that nurses became less capable of:(a)Synchronizing nursing rhythms;(b)Practicing with disease and/or condition specific familiarity;(c)Prioritizing each nursing task;(d)Predicting care requirements;(e)Practicing with inter-professional relationality as they were nursing the outlier patient.

Reconsidering the inherent nature of patient inclusiveness in nursing [26] and the concept of professional nursing accountability [28], nurses can be accused of negligence and breach of duty for inappropriate care [53]. The theory of nursing, as informed by the Standard of Practice, requires nurses to be responsible for “coordinate(ing) resources effectively and efficiently for planned actions; … for the planning and communication of nursing practice ... (and for ensuring) agreed plans are developed in partnership” [28]. In the case of John’s experience of delay care for outlier patients due to equipment unavailability, for example, he is at risk of violating the Standard of Practice. However, he may not violate the “prevailing” Standard of Practice, as “what may be considered negligent now may not have been considered negligent at the time” [54]. 

At the time when John was nursing the outlier patients, John coordinated care and liaised with other ward to get the equipment. The initial inquiry concerns whether he is capable of planning care ahead and getting the equipment early enough to avoid care lag. When care lag occurs, the outlier patient can bring a claim of malpractice against a nurse in the legal system, where malpractice refers to a form of negligence committed in carrying out professional duties [54,55]. The later inquiry then concerns whether John possesses specialized knowledge and skills to determine the correct timing for organizing the equipment. Considering John’s nursing knowledge and skills developed over time in the respiratory ward, his capabilities of nursing a respiratory patient may not match the specialized needs of the outlier patients. He may not possess all specialized knowledge and skills required for caring the outlier patients. When the nurse’s rungs in the care ladder are disrupted by the outlier patients, and he or she becomes unable to reach an appropriate level of care, the nurse is not competent in nursing the outlier patient. Nurses who choose to practice outside their scope of practice are placing themselves at risk for malpractice claims [55]. Whether the nurse passively accepting the work arrangement of nursing the outlier patients is accountable for the malpractice claims is controversial, for accountability refers to the nurses’ answer to “people in their care, the nursing regulatory authority, their employers and the public“ and to nurses being accountable for “their actions, behaviors and the responsibilities that are inherent in their nursing roles” [28]. 

### 5.2. Implications on Nursing Practice: Allevating Measures in Place, Studied Phenomenon Remains Prominent

At present, measures to alleviate the extra workload for nurses caring for outlier patients have been increasingly implemented in acute care hospitals. For instance, in relieving the frustration of nursing staff regarding contacting the medical team for the outlier patients, the job position of Medical outlier nursing coordinator has been created, and switchboard managers help to redirect calls from the medical-on-call team to the outlier-medical team [56]. Nevertheless, nurses who delegate aspects of their nursing practice to another person retain accountability for monitoring the practice’s outcomes [28]. This contrasts with the case of measuring patient’s outcome using a standardized pathway, in which the hospital system, instead of the nurse, is held accountable for the outcome variance caused by unavailable equipment, medications and supplies [57]. Future research directions in softening the concrete and redeveloping the Standard of Care and nurses’ accountability according to the flow of practice changes could be considered [54]. Furthermore, while a recent systematic review suggested no association between greater risk of malpractice liability and health care quality [58], the emphasis is on engaging in physicians’ voice rather than the nurses’ voice [59]. While the delayed nursing care and missed care remain prominent in acute care hospitals internationally [60,61], nursing outlier patients can be a potential stressor worsening the current situation. Increases in remaining hospital length of stay (22.8%) and in hospital readmission within 30 days (13.1%) for outlier patients have been reported when comparing to specialty-appropriate patients [62]. The compromised patient’s outcome indicates a need to investigate the association between malpractice liability and appropriate level of care in nursing the outlier patients.

Current research also attempts to minimize the health complications of outlier patients by identifying the high-risk groups, such as male outliers with the length of prolonged outlying stays greater than 2.5 days and classifying these higher risk individuals as ineligible to be outlier patients [63]. However, the occurrence of outlier patients and the studied phenomenon persists. Alternatively, measures to eliminate the occurrence of outlier patients, such as “dedicated configuration” that transfers patients to another hospital when the waiting time threshold for the matching hospital beds has been reached, have been implemented [62]. The feasibility of solely employing “dedicated configuration” is questionable given the global hospital beds shortage and the high hospital bed occupancy rate [16,17,18,19]. With organization change, some specialty wards are merging, and nurses with different specialties can learn from each other. “Clustered overflow configuration” is a reconstruction of hospital place, where the specialties are partitioned into an overflow ward surrounded by specialty wards [64]. Recent healthcare reform supports redesigning and modelling multi-specialty hospitals to promote patient-centered care [65,66]. However, these developments take time, which is often lacking in the real world. If the organization of hospital environments reverted to only generalist medical and surgical wards, could nurses cope with the vast ranges of knowledge and skills required? The history of specialization in medicine and of the hospitals designed to deliver the best patient care within those specializations, would suggest not. While literature claimed that outlier patients will be admitted to the overflow ward staffed with “multi-skilled nursing teams capable of caring for all specialties” [62], the inquiry concerns how to assure to the public that nurses in practice possess continuing competence, in other words, the ability to practice safe and quality nursing in the overflow ward [67]. Barriers for establishing the continuing competence, such as the lack of personal insight to reflect on or assess the nursing performance, the failure to seek continuing professional development opportunities and to recognize unsafe practice, ought to be considered carefully in the restructuring of hospital wards or bed management planning [68]. In terms of the nurse’s level, examples of operational interruptions include insecurity and dissatisfaction in newly merged hospital wards [69]. Extending Jameton’s definition of moral distress as the psychological distress that occurs “when one knows the right thing to do, but institutional constraints make it nearly impossible to pursue the right course of action” [70] (p. 6), nursing the outlier patient is the experience of a nurse nursing the outlier patient based on the ontological moral event and the associated psychological distress [71]. The ethical conflict in hospital policy caused by the extra time needed by the nurse to care for outlier patients and the cost-control required from the organization’s perspective has contributed to the moral distress experienced by nurses [72]. Care lags and missed nursing care have also been identified as global stressors for nurse job enjoyment and job retention [73]. 

### 5.3. Implications for Nursing Practice: Specialty Practice Being Challenged 

In summary, constraints for the nurses have been created by their external and internal worlds as they nurse outlier patients. Externally, the organizational designs of workplace and work have initially positioned the nurse in a familiar situation of nursing specialty appropriate patients. Nursing in this setting is experienced as repetitive tasks of adding or removing rungs from the composite care ladder to fit various individual patients’ needs. Clinical reasoning becomes “ritualize(d) experience” with manageable risks [49]. The nurse perceives herself as efficient and competent nursing on the care ladder between the surfaces of anchorage. The nurse normalized the use of the care ladder as her everyday nursing. As the external world of organization introduces the occurrence of outlier patients, some nurses become stuck in “the horizon of the past” at the comfort zone internally [38]. This means that they continue to use a composite care ladder that was developed within another specialty and resist attempts to add or remove rungs to become more flexible in practice. This specialty rigidity is strengthened by educational courses designed for enhancing specialty knowledge within current physiological systems orientations. These formal courses previously existed at all levels of the educational hierarchy in NSW, where the increasing complexities in patient care make it extremely difficult for nurses to maintain a generalist knowledge of all areas of nursing practice. Organizations requiring nurses to work with outlier patients are depending on the goodwill of nurses to focus on the relatively small amounts of completely generalizable nursing knowledge and skills while practicing in a safe manner. Thus, the registered nurse would check all unfamiliar medications and seek information before undertaking any unfamiliar procedures. However, any knowledge gained through this process would not be adequately rehearsed and therefore would not be retained. This is especially the case as the next outlier patient may come from a completely different specialization. 

In terms of the organization level, additional costs resulting from mixed and inconclusive impacts on quality of care during the immediate post-hospital merger and realignment period have questioned the feasibility of merging the wards [74]. The organizational expectation that nurses will be able to cope with any type of outlier patients undervalues the nurse’s experience, with accumulated knowledge and skills. It also depends on the goodwill of patients and families to be “coped with” rather that provided with an appropriate (even adequate) level of care. Nursing on the care ladder is initially a “ritualized experience” with manageable risks. In terms of the nurses’ experience, bounded in a specialty ward filled with specialty-appropriate patients, some nurses automatically undertake activities that provide appropriate care. The nurse does not need to reflect on the care ladder while she determines the capabilities required by each patient as these rungs are familiar and internalized. The nurse automatically and directly applies the composite care ladder to all specialty-appropriate patients and then adds or removes rungs from her composite care ladder to fit for the requirements by each specialty patient in her everyday nursing. The composite care ladder makes nursing specialty-appropriate patients manageable, as nurses save time and effort by not having to construct the care ladder from areas of knowledge outside their own experience. Participants in this study reported feeling disrespected by a hospital system that undervalued the time and effort they had spent in developing specific knowledge and skills while at the same time expecting them to accept the medico-legal risks of working with patients beyond their field of expertise. This high level of expectation is compounded by the overstretched resources of most acute care settings. Thus, the time of nurses was consumed chasing equipment that their ward did not have and medical staff with whom they were unfamiliar. 

Nursing outlier patients is an example that challenges specialty practice. Nursing specialization is inevitable with knowledge and technological advancement. The phenomenological orientation of a care ladder allows a deeper understanding of the phenomenon from the frontline nurse’s perspective and explores the nurse’s experience as the mechanisms of the work environment may negatively influence patient and nurse outcomes [71,72]. This phenomenological study highlighted that the nurse’s construct of specialty nursing is relevant to a particularly specialized work environment only. The advancement of knowledge and technology experienced by the nurse in one specialty may not be generalizable or transferrable across other specialties. The nurse’s capability is associated with the nurse’s experience of everyday nursing. Nursing outlier patients excludes the nurse from a normalized construct of nursing. The nurse therefore becomes less capable of nursing outlier patients, when compared to nursing specialty-appropriate patients. This study therefore suggests that nursing patients in specialty-appropriate wards may improve patient outcomes and alleviate the workload of frontline nurses. This paper further highlights the importance of promoting nurses’ perusal of continuing competencies, as well as initiating nursing leadership in the health policy domain [75]. Future directions for healthcare reform should place emphasis on building nurses’ competency and expertise for patient-centered care, rather than disrupting the nurses’ specialty constructs [76]. Implementation of expansion and enhancement of the nurses’ specialty constructs through mandatory education, competencies and skill demonstrations are recommended to minimize the effects of work environment on patient and nurse outcomes [77,78,79,80].

### 5.4. Limitations

The main limitation of this study is that the participants recruited are mainly nurses with many years of nursing experience rather than a diversity of nurses. For example, this study lacks the voices of nurses at their early stage of the nursing career. Most nurses recruited were above 30 years of age, with the years of nursing experience ranging from 7–38 years. Our findings illustrated that the unease and challenges of nursing the outlier patients are expressed by nurses with extensive years of experience, which is comparable to the earlier discussion on Benner’s discourse in the Background section [7]. Some participants reported that they could only provide basic care instead of the specialized care as required by the outlier patients. The findings alarmed the nurses, the patients, the bed managers and the organization, considering the influence of workplace organization on nursing experience. Future studies targeting participant recruitment for nurses at the beginning of their career may capture and optimize the diversity of the nurses’ experience in the studied phenomenon, hence enhancing collection of richer data. 

Other limitations, such as the interviews varying in length and the potential participant fatigue poses accuracy risk. The duration of interviews in this study varies from thirty minutes to one and a half hours. Participants may experience interview fatigue when a single interview lasts from one to two hours [81]. Setting up a new appointment for another occasion may be recommended for future studies [81]. 

For the purpose of assuring transcription accuracy, the first author listened to the audio file and compared it to the transcript, while checking any spelling or transcription errors. Utterances are represented and silences are documented. The second and third authors have also randomly listened to some of the audio files and assured that it was transcribed completely [82].

My immersion as a nurse may be regarded as a potential to influence the analytical distance required as a researcher. Nevertheless. I see this involvement as positive and important in informing my interpretation of the phenomenon. Through reflecting on my experience, I recognized that my experience of nursing outlier patients is a part of the multiple realities. As discussed previously in Table 2, memos were written after each interview in order to demonstrate reflexivity [83]. I would argue that my reflexivity enhances the reader’s understanding of my own values and therefore adds confirmability and validity to this study [84,85].

Critique of data collected as being out-of-date and the risk of developing bias over time have been considered. Phenomenological reflection is retrospective in nature. While data collected dated back to 2009–2011, phenomenological interpretation does not limit itself to the time of data collection. Phenomenological interpretation continues to be interpreted by authors at present and by readers in the future. As stated by van Manen, “Phenomenological research is the attentive practice of thoughtfulness” [86] (p .38).

At present, the studied phenomenon remains prominent despite measures for reducing the extra workload for nurses caring for outlier patients being in place. Rather than regarding the data analysis as bias over time, the data collected from 2009–2011 is part of the evolutionary trajectory of the studied phenomenon and remains relevant to current practice. For the same event, the person may have different interpretations or meanings attached to a particular experience on different horizons of time. Rather than chasing the time and timeliness in appraising clinical relevance, this manuscript is aiming for the restructure of temporality. “Past data” as described in this paper can facilitate understanding of the studied phenomenon and opens possibilities for new understanding of this dynamic phenomena [87], as well as serving as a future reference in areas where there is a lack of research [88]. It is also important to acknowledge the impossibility to fully and absolutely understand a phenomenon [89].

## 6. Conclusions

The phenomenon of nursing outlier patients is an example of the frontline nurses’ experience being influenced by the organization of work. The good intention of facilitating patient flow by the hospital bed management has resulted in the misfit between patients’ needs and nurses’ specialty construct. Nursing out of scope of practice has potential implications on the nurses’ accountability and liability for negligence. With the current organization’s strategies for restructuring the ward and increasing manpower to facilitate the care of outlier patients, the nurse dispensing care remains accountable. Emphasizing the nurse’s development of specialty knowledge and skills over time, there is a need to undertake further research to promote the importance of attaining continuing competency and to investigate the transferability of knowledge and experience across different specialized environments. The invisible work and the extra workload of nursing outlier patients when compared with nursing specialty-appropriate patients’ needs to be acknowledged with supportive measures to safeguard the level of care for all patients.

## Figures and Tables

**Figure 1 ijerph-17-05232-f001:**
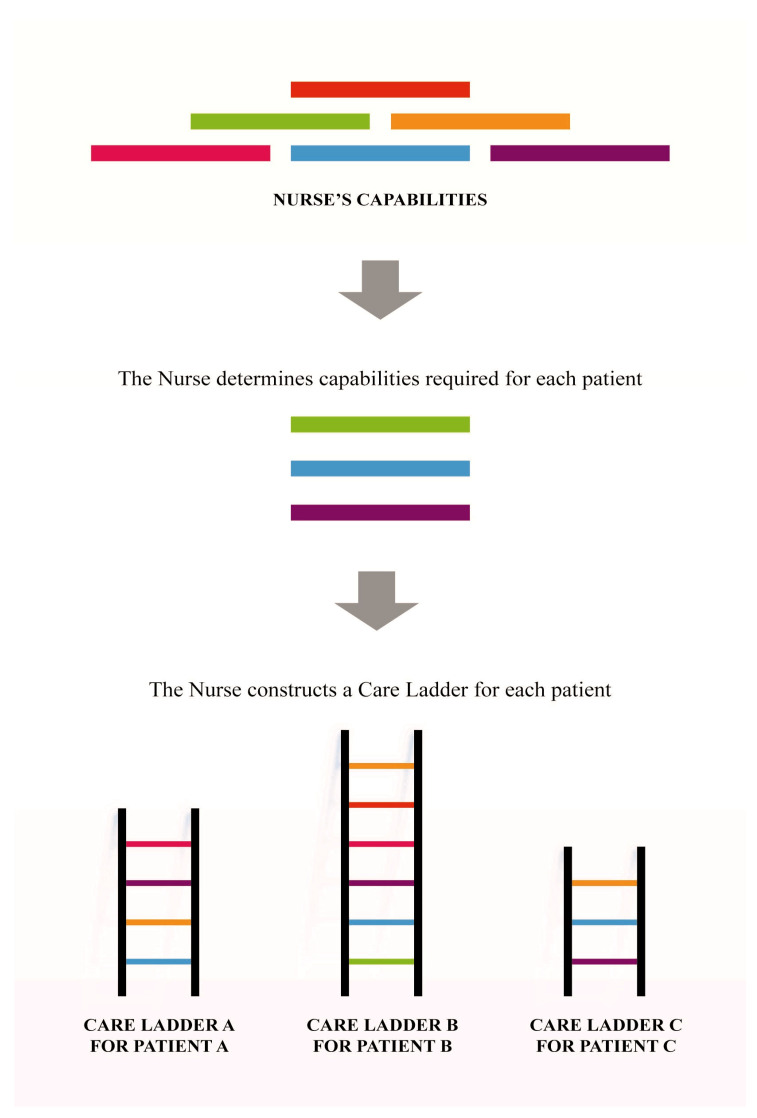
Distinctiveness of the nurse’s care ladder.

**Figure 2 ijerph-17-05232-f002:**
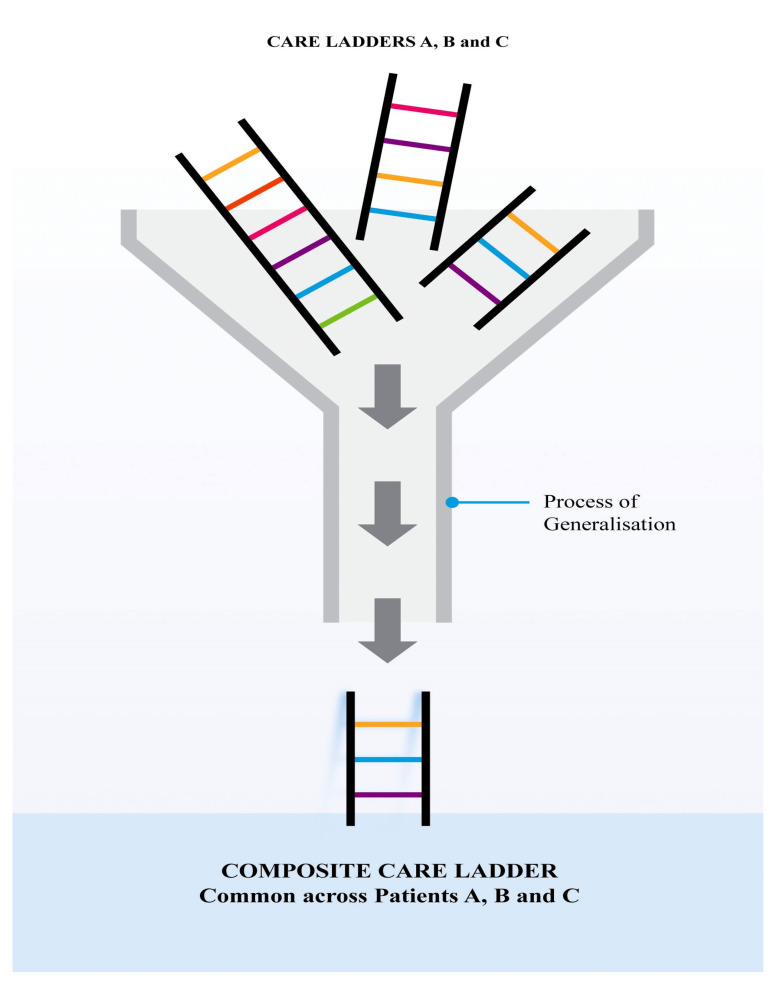
Construction of the composite care ladder from a multiplicity of care ladders.

**Figure 3 ijerph-17-05232-f003:**
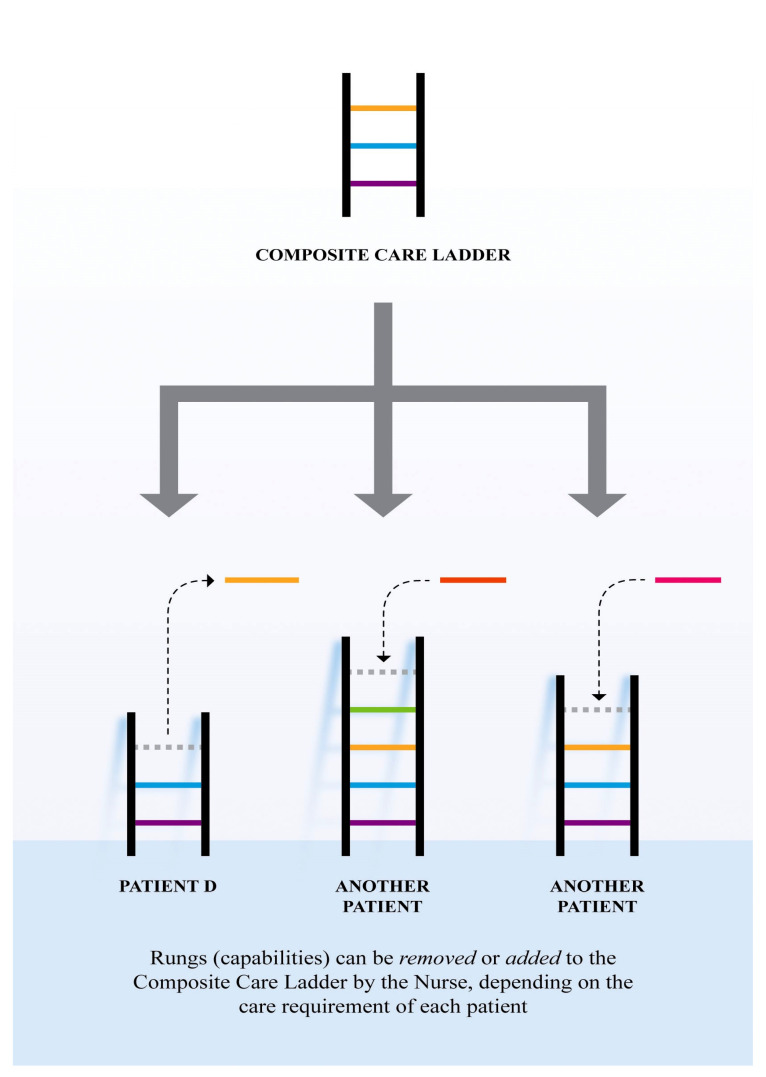
Composite care ladder.

**Figure 4 ijerph-17-05232-f004:**
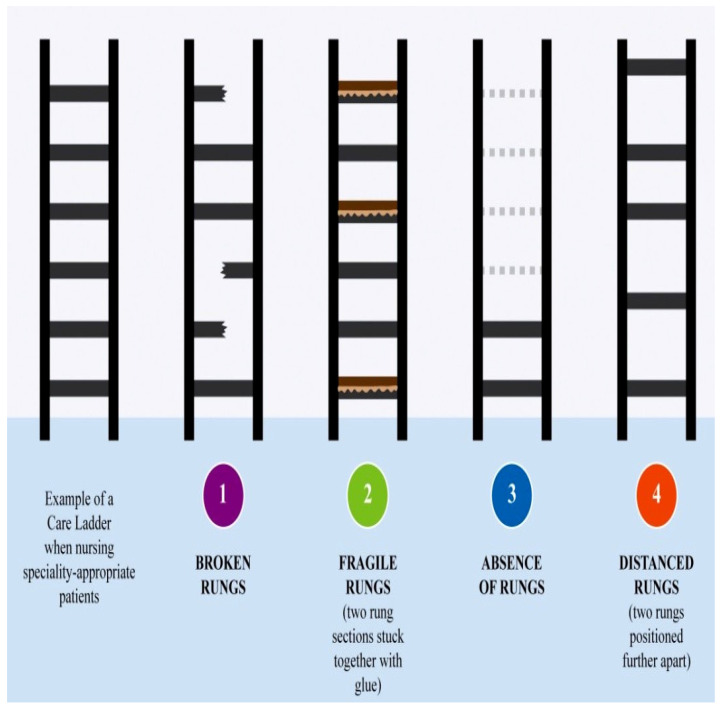
Four potential rung disruptions as an outcome of nursing outlier patients compared with nursing specialty-appropriate patients.

**Table 1 ijerph-17-05232-t001:** Demographic characteristics of the participants.

Pseudonym	Gender	Age by Decade	Years of Nursing Experience	Area of Practice
Rainbow	F	50’s	12	Coronary Care Unit
Agnes	F	50’s	31	Rehabilitation/Acute Stroke Unit
Ann	F	60’s	>30	Discharge planning
John	M	50’s	38	Respiratory ward
Madeline	F	40’s	23	Respiratory ward
Marie	F	30’s	17	Medical ward
Peter	M	50’s	>30	Cardiac ward
Claire	F	50’s	34	Gynecology ward
Mary	F	30’s	7	Transplant ward
Hope	F	50’s	32	Neurology
Kay	F	50’s	38	Pediatric

**Table 2 ijerph-17-05232-t002:** The non-linear process of phenomenological data analysis applied in this study.

Research Phases	Phenomenological Data Analysis
1. During the interview	Listen to participant’s description of their experience of nursing the outlier patients Develop first interpretation of what participants have said
2. Following each interview session	Write summary for each interview session Record reflective notes (memoing) according to my experience as a nurse and as a researcher Record immediate apparent concepts and themes (if any)
3. During transcription	Use my personal experience as a starting point for data interpretation Record my later understanding of what participants have said
4. During the line-by-line analysis with interview scripts	Discover the commonalities and differences among subthemes emerged Uncover initial thematic aspect by referring to the summaries and reflection written previously

**Table 3 ijerph-17-05232-t003:** Conducting initial thematic analysis.

Initial Themes	Initial Sub-Themes
1. Nurse’s reported feeling from experience of nursing the outlier patients	In Doubt; Belittled; Hesitate; Fearful; Dissatisfied; (Learnt) Helplessness; In Doubt; Devastating; Acceptance/Indifference; Uncertainty; Frustrated re: patient care/outcome; Frustrated re: Uncertainty; Frustrated re: pressure; Frustrated re: lack of support/uncaring attitude; Being ignored; Frustrated re: staff and resources; Stress re: Lack of support; Frustrated; Not confident; Abuse; Stress re: intense workload; Not prepared; Guilt; Difficult; Bad; Unfamiliar; Feeling stress; Limited; Failure; Feeling hard; Not easy; Tense inside; Not welcoming/Don’t like/lack of interest; Not positive; Odd/Different; Stress: patient not getting care; Worry; Frustrated; Inadequate; Inappropriate; Anger re: uncaring attitude; Painful; Awful; Disappointed/unhappy; Unease
2. Perceived care ladder	Basic-Optimal/good; Basic-Human; Basic- Continuity; Something missing; Basic- Comprehensive; Generic- Specialized; Basic- Adequate- Specialized-Appropriate
3. Perceived care of outlier patients	Minimal care; Missing in care; Not basic care; Inappropriate care/inadequate; Compromised care; Best possible nursing; No predictability; not good nursing care/only basics; Not continuity of care; Not just babysitting/require intensive look after

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
