# Peer review of "The Frontline Nurse’s Experience of Nursing Outlier Patients"

_ijerph, 2020, doi:10.3390/ijerph17145232_

Round 1
Reviewer 1 Report
The introduction raises paragraphs of difficult understanding for the non-health public. Contextualization should be improved.
Clearly explained methods and designs.
Also, it results in present images that help to understand.
Robust discussion, nevertheless it is hindered by the number of references with more than five years (only 60% from 2015 to 2020).
Author Response
Thank you for considering my submission the revised manuscript of ijerph-852471, titled “The frontline nurse’s experience of caring the outlier patients”. I sincerely appreciate your constructive comments and suggestions, which helped to improve the manuscript. We have gone through all comments and suggestions, and revised the Manuscript accordingly. Revisions have been highlighted using Track-changes function in Microsoft word.
I am committed to have my first research paper published and any further comments and suggestions will be gratefully received. Please see attached my responses to the review report. Thanks.
Title: The frontline nurse’s experience of nursing the outlier patients
Author’s reply to the Review Report from Reviewer 1
|
Reviewer 1’s Comments |
Author’s Response/Revisions |
|
- The introduction raises paragraphs of difficult understanding for the non-health public. Contextualization should be improved.
|
Thank you for this suggestion. Title have been revised as “The frontline nurse’s experience of nursing the outlier patients” as listed on p.1 line 1-3.
|
|
- Clearly explained methods and designs. Also, it results in present images that help to understand. |
Thank you for your encouraging comment.
|
|
- Robust discussion, nevertheless it is hindered by the number of references with more than five years (only 60% from 2015 to 2020). |
Thank you for the valuable comment. We appreciate you for observing this. We acknowledge the importance of having the references less than five years.
Apart from seminal works from philosophers, we have carefully considered the content of a citation if it is more than five years. We also ensure that we have covered the most current research in this field. In this revised manuscript, more references within the last five years have been added. The opportunity for considering our attempt in publishing this manuscript will be gratefully appreciated. (Please see p.19 line 1116-1121, and p. 22-23 line 1279-1298 for newly added references)
|
Reviewer 2 Report
Thank you for the opportunity to review this paper. This paper sought to report findings from a study that explored nurses' perceptions of care provided to "outlier" patients using hermeneutic phenomenological methods. Outlier patients, per the implied definition in the paper, are patients who are placed in a nursing unit due to bed availability and perhaps not consistent with the patient type typically seen on that unit. The following could strengthen the paper. The title of the paper doesn't read very clearly. The first part is the best title with modification "The frontline nurses' experience of caring for the outlier patient".
The concept of the outlier patient isn't a new phenomenon. I can recall being assigned patients who were outside my population expertise. As the paper states, we as nurses dive in and provide care without question. One could argue that the outlier patient could also be one that requires skills or knowledge that the nurse has either not before experienced, such as the rare disease process or applied equipment. Which then brings into the argument of novice to expert as developed by Brenner. I am surprised that this didn't not appear as background in the paper to help set up the argument and then let the reader know the focus is the outlier patient on the unit that does not fit the typical patient population.
The inclusion and exclusion criteria appears to be very broad, capturing those who worked in a hospital within two years. Years of experience, degree obtained, license, full time/part time, float elements are unclear. The participants were all well experienced nurses, seven years the least and >30 to 38 the highest. The implication from the inclusion criteria and recruitment is that this was a coincidence. I think having captured inexperienced nurses and their perspectives would have been much richer data. For this reason, this should be explained and acknowledged that the sample was overwhelmingly well experienced.
Results-If I understand this correctly, one participant mentioned a ladder and that led to the model? Here I was looking for the collection of categories and supported exemplars to support the conclusions. The presentation of the ladder model and analogies with the rungs is nice, however, it is unclear how this fits with the respondents. Reading onward, and reading the categories such as 3.1 with synchronizing and so forth, I didn't understand the link to the ladder. 3.1 onward provided great insights into the responses and understanding of caring for a patient that is an outlier for expert nurses as defined by a patient who doesn't fit the unit specialization.
What was the process for achieving agreement with the interpretation of the data? Was the transcription verified for accuracy? Interviews varied in length and some were very long, questioning participant fatigue. What were the limitations of this study? These are unclear in the paper.
If the ladder model came from the interviews in total, then it could be moved to the end of the results section. Otherwise, this remains unclear.
The discussion presents the 5 categories and these are clear and make sense with the type of data.
Author Response
Thank you for considering my submission the revised manuscript of ijerph-852471, titled “The frontline nurse’s experience of caring the outlier patients”. I sincerely appreciate your constructive comments and suggestions, which helped to improve the manuscript. We have gone through all comments and suggestions, and revised the Manuscript accordingly. Revisions have been highlighted using Track-changes function in Microsoft word.
I am committed to have my first research paper published and any further comments and suggestions will be gratefully received. Please see attached my responses to the review report. Thanks.
Title: The frontline nurse’s experience of nursing the outlier patients
Author’s reply to the Review Report from Reviewer 2
|
Reviewer 2’s Comments |
Author’s Response/Revisions |
|
- The title of the paper doesn't read very clearly. The first part is the best title with modification "The frontline nurses' experience of caring for the outlier patient". |
Thank you for a very reasonable comment. Title have been re-phrased as “The frontline nurse’s experience of caring the outlier patients” as listed on p.1 line 1-3.
|
|
- The concept of the outlier patient isn't a new phenomenon. I can recall being assigned patients who were outside my population expertise. As the paper states, we as nurses dive in and provide care without question.
One could argue that the outlier patient could also be one that requires skills or knowledge that the nurse has either not before experienced, such as the rare disease process or applied equipment. Which then brings into the argument of novice to expert as developed by Brenner.
I am surprised that this didn't not appear as background in the paper to help set up the argument and then let the reader know the focus is the outlier patient on the unit that does not fit the typical patient population.
|
Thank you for sharing your experience and for this important detail. The discussion of novice to expert as developed by Benner and the experience of nursing patients with rare disease process or applied equipment have been added in the background section on p.2 line 59-74.
|
|
- The inclusion and exclusion criteria appears to be very broad, capturing those who worked in a hospital within two years. Years of experience, degree obtained, license, full time/part time, float elements are unclear.
The participants were all well experienced nurses, seven years the least and >30 to 38 the highest. The implication from the inclusion criteria and recruitment is that this was a coincidence.
Most nurses recruited were above 30s year of age, with the least experience of seven years and the most experience of 38 years I think having captured inexperienced nurses and their perspectives would have been much richer data. For this reason, this should be explained and acknowledged that the sample was overwhelmingly well experienced. |
Thank you for the valuable comment. Limitations have been added on p.17 line 944-955 to acknowledge the actually recruited nurses with extensive years of experiences, as compared to the initial attempt of including diversity of nurses.
|
|
Results - If I understand this correctly, one participant mentioned a ladder and that led to the model? Here I was looking for the collection of categories and supported exemplars to support the conclusions. The presentation of the ladder model and analogies with the rungs is nice, however, it is unclear how this fits with the respondents.
|
Thank you for supporting the clarity of our manuscript. Clarification of the ladder model emerging from participants’ excerpts have been added on p.7 line 448-471.
|
|
- Reading onward, and reading the categories such as 3.1 with synchronizing and so forth, I didn't understand the link to the ladder.
|
Thank you for this clarifying comment.
Categories 3.1- 3.4 (Currently being modified as 4.1.-4.4) are considered as the outcome of nursing the outlier patients, which consequentially lead to later discussion of the “four potential rung disruptions” as shown in Figure 4 on p.10. (Please see p.11 line 552 for clarification.)
|
|
-3.1 onward provided great insights into the responses and understanding of caring for a patient that is an outlier for expert nurses as defined by a patient who doesn't fit the unit specialization. |
Thank you for your encouraging comments. |
|
- What was the process for achieving agreement with the interpretation of the data? - Was the transcription verified for accuracy?
|
Thank you for your comment. Measure have been taken to maximize the accuracy of the data. For instance, constant comparisons have been made between the first author’s analysis and the independent analysis conducted by second and third authors, as mentioned on p.5 line 406-407.
The discussion regarding the verified transcription accuracy have been added on p.18 line 1023-1026. |
|
- Interviews varied in length and some were very long, questioning participant fatigue. What were the limitations of this study? These are unclear in the paper. |
Great suggestion, thank you. Discussion of participant fatigue have been added in the section of future implications and limitations on p.17-18 line 956-1032.
|
|
- If the ladder model came from the interviews in total, then it could be moved to the end of the results section. Otherwise, this remains unclear.
|
Thank you for your suggestion. Instead of moving this discussion to the end of the results sections, we have introduced the care ladder as the phenomenological orientation in earlier result sections to facilitate later discussion in the results section.
|
|
- The discussion presents the 5 categories and these are clear and make sense with the type of data. |
Thank you for your nice comments. |

Reviewer 3 Report
Dear authors,
The work is of great interest in the field of nursing practice and the consequences of treating atypical patient cases. It has great virtues but also some weaknesses. These are the detected ones:
- Section 1 should divide it into 2: Introduction and another that encompasses the theoretical background that you propose.
- The question or problem that you have detected to carry out the investigation does not appear well formulated.
- It would be good if you graphically present the steps of the applied methodology.
- The data collected is from 2009 to 2011. May there be any bias over time? If you believe that this is not the case, they should clarify it, and point out that they do not have current data. Past data may give inconsistent results at a later time.
- The figures have poor quality, they are blurred. Improve them.
- The discussion section is very long to read. Divide it in relation to the results you discuss in section 3.
- What limitations have you found when carrying out the study? Express them in the text.
- The references are not expressed as indicated by the journal. Take care of those details.
Author Response
Thank you for considering my submission the revised manuscript of ijerph-852471, titled “The frontline nurse’s experience of caring the outlier patients”. I sincerely appreciate your constructive comments and suggestions, which helped to improve the manuscript. We have gone through all comments and suggestions, and revised the Manuscript accordingly. Revisions have been highlighted using Track-changes function in Microsoft word.
I am committed to have my first research paper published and any further comments and suggestions will be gratefully received. Please see attached my responses to the review report. Thanks.
Title: The frontline nurse’s experience of nursing the outlier patients
Author’s reply to the Review Report from Reviewer 3
|
Reviewer 2’s Comments |
Author’s Response/Revisions |
|
- Section 1 should divide it into 2: Introduction and another that encompasses the theoretical background that you propose. |
Thank you for your comment. The paragraphs have been divided into Introduction and Background sections as listed on p.1 line 32-42 and p.2-4 line 50-305. |
|
- The question or problem that you have detected to carry out the investigation does not appear well formulated. |
Thank you. We have elaborated on the background section (on p.2-4 line 50-305) to facilitate better formulation of the research problem. |
|
- It would be good if you graphically present the steps of the applied methodology. |
Thank you for your suggestion. Table 2 and Table 3 have been added on p.5 line 414- 427 to facilitate understanding about the phenomenological data analysis applied in this study.
|
|
- The data collected is from 2009 to 2011. May there be any bias over time? If you believe that this is not the case, they should clarify it, and point out that they do not have current data. Past data may give inconsistent results at a later time. |
Thank you for the important comment. Discussion of possible bias over time have been added as limitation, as listed on p.18 line 1033-1060. The opportunity for considering our attempt in publishing the data will be gratefully appreciated. |
|
- The figures have poor quality, they are blurred. Improve them. |
Thank you for your observation. The sharpness of the figures 1-4 on p.8-11 have now been adjusted. |
|
- The discussion section is very long to read. Divide it in relation to the results you discuss in section 3. |
Thank you for your suggestion. Subheadings have been added to divide the results into sections, as listed on p. 14 line 702 to p.18 line 1060.
The subheadings include: 5.1. Implications on nursing practice: Organizational commitment for nurses and the associated liability and accountability 5.2. Implications on nursing practice: Alleviating measures in place, studied phenomenon remains prominent 5.3. Implications on nursing practice: Specialty practice being challenged 5.4. Future implications and limitations
|
|
- What limitations have you found when carrying out the study? Express them in the text. |
Thank you for observing this. Discussion of limitations have been added on p. 17-18 line 944- 1050. |
|
- The references are not expressed as indicated by the journal. Take care of those details. |
We have revised the manuscript thoroughly regarding the referencing style.
|
Round 2
Reviewer 2 Report
Thank you for the opportunity to review this revised manuscript. The suggestions recommended and comments put forth have been addressed. The paper reads better and the logic is more in sync. Thanks.
Author Response
Thank you again for your positive comments for our manuscript of ijerph-852471, titled “The frontline nurse’s experience of caring the outlier patients”. Your comments have improved and strengthened our manuscript.
Reviewer 3 Report
Dear authors,
I gratefully check the improvement of the manuscript, although there are still details to correct:
- Figures 1 to 4: too blurry
- You need to read the authors' guideline to avoid submitting a manuscript as requested by the journal (tables, words in bold ...).
- They used the word "study" 43 times. Too many. Maybe some synonym?
Author Response
Please see the attachment.

This manuscript is a resubmission of an earlier submission. The following is a list of the peer review reports and author responses from that submission.
Round 1
Reviewer 1 Report
Dear authors,
I congratulate you on your work. This presents great advantages and opportunities, although I also observe some doubts that I let you know to improve it:
- The title is confusing, even more so the subtitle. It should be more concise.
- Write more keywords. Think of these as a search tool.
- Line 39: “unsaid” should be in quotation marks and in italics?
- The first paragraph only has a reference? Is everything written from your previous knowledge?
- The Introduction should be a separate section from the subsections they do. These should go in another section.
- The objective of the study is not clearly defined, nor how they would define it.
- Why is this study important in the area of health? Why is it necessary to do it?
- The conclusions are too short. If they are not going to extend them, join them to the previous section.
- Anyway, the conclusion should be more rigorous and more extensive, after approaching this work. It is not a phrase that you say to a friend in an elevator, but to define what they have extracted from this work and why it is interesting.
Reviewer 2 Report
Dear authors
The conclusion is too short and biased by the author's personal interpretation. It is recommended to strengthen their interpretation of results with some quantitative method that evaluates the professional performance of specialized nurses in this type of high-demand work environment.